# T2V2: A Unified Non-Autoregressive Model for Speech Recognition and Synthesis via Multitask Learning

**Nabarun Goswami**[1], **Hanqin Wang**[1], **Tatsuya Harada**[1,2]
[1]The University of Tokyo, Japan
[2]RIKEN, Japan
{nabarungoswami, wang, harada}@mi.t.u-tokyo.ac.jp

## Abstract

We introduce T2V2 (**T**ext to **V**oice and **V**oice to **T**ext), a unified non-autoregressive model capable of performing both automatic speech recognition (ASR) and text-to-speech (TTS) synthesis within the same framework. T2V2 uses a shared Conformer backbone with rotary positional embeddings to efficiently handle these core tasks, with ASR trained using Connectionist Temporal Classification (CTC) loss and TTS using masked language modeling (MLM) loss. The model operates on discrete tokens, where speech tokens are generated by clustering features from a self-supervised learning model. To further enhance performance, we introduce auxiliary tasks: CTC error correction to refine raw ASR outputs using contextual information from speech embeddings, and unconditional speech MLM, enabling classifier free guidance to improve TTS. Our method is self-contained, leveraging intermediate CTC outputs to align text and speech using Monotonic Alignment Search, without relying on external aligners. We perform extensive experimental evaluation to verify the efficacy of the T2V2 framework, achieving state-of-the-art performance on TTS task and competitive performance in discrete ASR.

## 1 Introduction

Speech recognition and synthesis are foundational tasks in human-computer interaction, enabling applications ranging from voice assistants to automated transcription services. Traditionally, both tasks rely heavily on autoregressive models, which generate sequences one token at a time, resulting in higher latency and limited efficiency (Graves, 2012; Sutskever et al., 2014; Chan et al., 2016; Wang et al., 2017; Li et al., 2019; Casanova et al., 2024). Recently, non-autoregressive models have gained traction by reducing inference latency while maintaining robustness in both TTS and ASR(Ren et al., 2020; Kim et al., 2021; Lee et al., 2023; Casanova et al., 2022; Graves et al., 2006; Higuchi et al., 2020; 2021a). Another promising approach is the use of discrete speech tokens, made possible by advances in neural audio codecs(Zeghidour et al., 2021; Kumar et al., 2024) (*acoustic tokens*) and clustering features from large-scale self-supervised speech models(Hsu et al., 2021; Baevski et al., 2020) (*content tokens*). These discrete tokens offer advantages in sequence modeling, particularly when paired with transformer-like architectures. Discrete tokens enable efficient storage and transmission, ideal for large datasets. They also serve as intermediate representations that capture acoustic and linguistic information while being less speaker-specific, aligning well with the unified modality in our framework.

Despite the recent success of non-autoregressive models in both ASR and TTS, integrating these tasks into a unified framework remains a significant challenge. A few works (Rubenstein et al., 2023; Wang et al., 2024; Maiti et al., 2024; Yang et al., 2024a; Toyin et al., 2024) have explored a unified modeling of speech-text related tasks, but they are limited to AR encoder-decoder or decoder only methods. However, existing NAR models typically handle ASR and TTS separately, missing opportunities to share representations and leverage joint training to improve performance across tasks. While multitask learning offers potential for parameter sharing, the performance of individual tasks might suffer due to the inherent complexity of training multiple tasks simultaneously. Au-

thors in Xiujuan & Zhongke (2004), detail these challenges and common optimization approaches like scalarization and Pareto optimality. Another key challenge in NAR CTC-based ASR is the conditional independence assumption of CTC loss(Graves et al., 2006), which can lead to errors in transcription. Several works have tackled this problem with varying degrees of success(Higuchi et al., 2020; 2022; Chi et al., 2021; Nozaki & Komatsu, 2021). Moreover, NAR TTS models rely heavily on accurate alignment between text and speech, making external alignment tools(McAuliffe et al., 2017) necessary in most systems. A unified approach that can address both the independence limitations of CTC in ASR and the alignment challenges in TTS is needed. Last but not least, the use of discrete speech tokens, which have shown success in TTS (Borsos et al., 2023a; Rubenstein et al., 2023; Wang et al., 2023; Casanova et al., 2024) , has not been fully explored in ASR(Chang et al., 2023b; Yang et al., 2024b), leaving a gap in developing efficient models that can handle both tasks with the same set of representations. Addressing these challenges, while reducing reliance on external aligners and maintaining competitive performance, forms the core motivation of our work.

To address these challenges, we propose T2V2, a non-autoregressive model that unifies ASR and TTS using shared representations. T2V2 models *content tokens* with CTC-based training for ASR and a conditional masked language modeling approach for TTS, converting these tokens to *acoustic tokens* and ultimately to speech via the SoundStorm(Borsos et al., 2023b) and codec decoder. The core of T2V2 is a Conformer architecture with rotary positional embeddings (RoPE), capturing both local and long-range dependencies in speech and text. For TTS, Monotonic Alignment Search (MAS) is applied to intermediate CTC outputs, eliminating the need for external alignment tools and ensuring consistency across tasks.

To improve speech generation, we employ classifier-free guidance(Ho & Salimans, 2021), using an unconditional masked speech model to iteratively refine outputs. For ASR, we address CTC's independence limitation through a CTC error correction task, refining outputs with speech embeddings to recover time-step dependencies and enhance transcription accuracy. T2V2 efficiently handles both tasks within a unified framework, with auxiliary tasks like CTC error correction and unconditional speech MLM boosting performance. By leveraging discrete content tokens, our method improves sequence modeling and unifies speech and text processing. Our key contributions are as follows:

1. We propose T2V2, the first, to the best of our knowledge, unified NAR model for ASR and TTS and operates on discrete speech tokens derived from self-supervised models.

2. We leverage Monotonic Alignment Search (MAS) with intermediate CTC outputs for TTS alignment without relying on external tools for self-contained text-speech alignment.

3. We introduce a CTC error correction formulation to refine raw CTC outputs, improving ASR performance within the unified framework.

4. Extensive experimental validation confirms that T2V2 markedly improves synthesized speech robustness achieving state-of-the-art TTS performance and competitive discrete ASR performance.

## 2 MULTITASK T2V2 FRAMEWORK

The overview of our method is shown in Figure 1. The following sections provide a detailed description of each component and task.

### 2.1 NOTATION AND TERMINOLOGY

Here we introduce the notation used throughout the paper. The inputs to our model are discrete speech content tokens, denoted by $\boldsymbol{S} = \{s_1, s_2, \ldots, s_N\}$, where $N$ is the number of tokens in the speech sequence, and corresponding text tokens, represented as $\boldsymbol{T} = \{x_1, x_2, \ldots, x_L\}$, with $L$ being the text sequence length. These tokens are first converted into embeddings through their respective embedding layers, resulting in $\boldsymbol{Z}_S$ for speech embeddings and $\boldsymbol{Z}_T$ for text embeddings.

During training, subsets of tokens are masked depending on the task, and the masked tokens, along with their corresponding embeddings, are denoted with a superscript $M$, such as $\boldsymbol{S}^M$ and $\boldsymbol{Z}_S^M$. The task-specific masking schemes are detailed in Section 2.5.

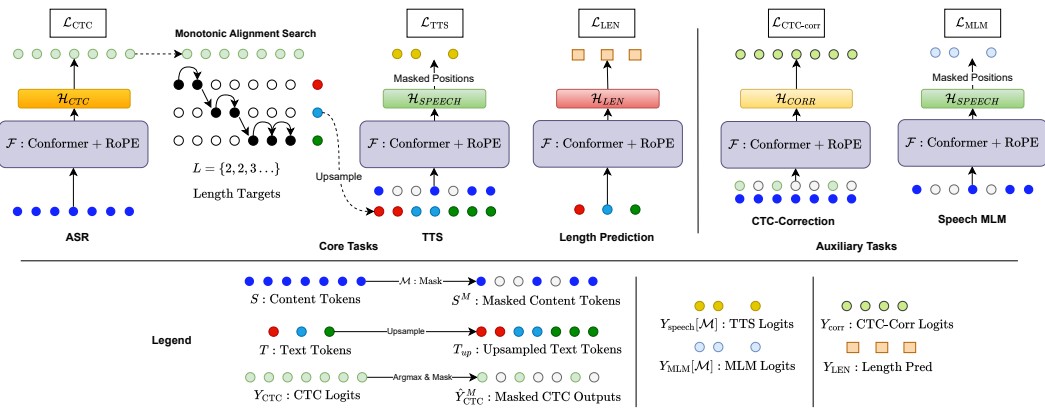

Figure 1: Overview of T2V2 architecture, best viewed in color. The parameters of the Conformer are shared across all tasks. The task specific heads with same color also share their parameters.

The shared Conformer encoder backbone is represented as $\mathcal{F}$, and the task-specific heads include the CTC prediction head, $\mathcal{H}_{\text{CTC}}$, CTC correction head, $\mathcal{H}_{\text{CORR}}$, speech prediction head, $\mathcal{H}_{\text{SPEECH}}$, and the length prediction head, $\mathcal{H}_{\text{LEN}}$.

## 2.2 CONFORMER WITH RoPE BACKBONE

In our proposed method, all tasks share a single backbone, $\mathcal{F}$, based on the Conformer (Gulati et al., 2020) architecture with rotary positional embeddings (RoPE) (Su et al., 2024). The Conformer combines convolutional layers, which capture local dependencies in the input sequences, with self-attention layers, which model long-range dependencies in both speech and text sequences.

Rotary Positional Embeddings (RoPE) further enhance the model's ability to process variable-length sequences by encoding positional information directly within the self-attention mechanism. Unlike traditional positional encodings, RoPE handles long sequences more effectively, making it especially beneficial for speech and text tasks that require both fine-grained temporal modeling and global context understanding. This combination has been demonstrated to be particularly effective for speech recognition (Li et al., 2021) and synthesis (Borsos et al., 2023b) tasks. Thus we utilize this combination as the backbone to efficiently share parameters across multiple tasks, improving both performance and flexibility in multitask learning.

## 2.3 CORE TASKS

### 2.3.1 SPEECH RECOGNITION WITH CTC

For the speech recognition task, the speech embeddings $\boldsymbol{Z}_S$, are passed through the shared Conformer backbone $\mathcal{F}$, and then through the CTC text prediction head $\mathcal{H}_{\text{CTC}}$. The Connectionist Temporal Classification (CTC) framework (Graves et al., 2006) is used to align the input speech with the target transcription without requiring explicit frame-level alignment. The predicted output logits over the vocabulary at each time step are given by:

$$\boldsymbol{Y}_{\text{CTC}} = \mathcal{H}_{\text{CTC}}(\mathcal{F}(\boldsymbol{Z}_S)), \tag{1}$$

where $\boldsymbol{Y}_{\text{CTC}}$ represents the CTC logits.

The CTC loss $\mathcal{L}_{\text{CTC}}$, is computed by marginalizing over all possible alignments between the input sequence and the target transcription $\boldsymbol{T}$:

$$\mathcal{L}_{\text{CTC}} = -\log \sum_{\mathbf{a} \in \mathcal{A}(\boldsymbol{T})} P(\mathbf{a}|\boldsymbol{Y}_{\text{CTC}}), \tag{2}$$

where $\mathcal{A}(\boldsymbol{T})$ is the set of all valid alignments of the target sequence $\boldsymbol{T}$. This formulation allows the model to predict the text sequence while accounting for varying lengths between the input speech and the target transcription.

### 2.3.2 TEXT TO SPEECH AND LENGTH PREDICTION

For the text-to-speech (TTS) task, we compute the alignment score matrix $M \in \mathbb{R}^{N \times L}$ by extracting the log probabilities of the text tokens $T$ from the CTC logits $Y_{\text{CTC}} \in \mathbb{R}^{N \times V}$ as:

$$M_{n,\ell} = Y_{\text{CTC},n,x_\ell}, \quad \forall n \in \{1, \ldots, N\}, \forall \ell \in \{1, \ldots, L\}. \tag{3}$$

Next, we apply the Monotonic Alignment Search (MAS) (Kim et al., 2020) to $M$ to obtain the alignment matrix $A \in \{0,1\}^{N \times L}$. The MAS establishes a monotonic mapping between the text tokens and the speech time steps. For details, we refer the reader to Kim et al. (2020).

The input to the Conformer backbone is the point-wise addition of the embeddings of the upsampled text tokens $T_{\text{up}} = \text{Upsample}(T, A)$, $Z_T^{\text{up}}$ and the masked speech embeddings $Z_S^M$: $X_{\text{TTS}} = Z_T^{\text{up}} + Z_S^M$. The backbone output is passed through the speech prediction head $\mathcal{H}_{\text{SPEECH}}$, and the loss is computed using cross-entropy focused on the masked positions:

$$\mathcal{L}_{\text{TTS}} = -\sum_{i \in \mathcal{M}} S_i \log P(\mathcal{H}_{\text{SPEECH}}(\mathcal{F}(X_{\text{TTS}}))_i), \tag{4}$$

where $\mathcal{M}$ is the set of masked positions, $S$ represents the target speech tokens, and $P(\cdot)$ is the predicted probability.

For the length prediction task, the target token lengths $L_i$ are derived from the MAS alignment matrix $A$. Text tokens are passed through the length prediction head $\mathcal{H}_{\text{LEN}}$, and the loss is computed using an L1 objective:

$$\mathcal{L}_{\text{LEN}} = \sum_i |\mathcal{H}_{\text{LEN}}(\mathcal{F}(T))_i - \log(L_i)| . \tag{5}$$

## 2.4 AUXILIARY TASKS

### 2.4.1 CTC ERROR CORRECTION

In traditional CTC-based models, outputs at each time step are conditionally independent (Graves et al., 2006), limiting their ability to capture long-term dependencies and recover from early errors. To address this, we introduce a CTC error correction task that refines initial CTC predictions using confidence based masked intermediate CTC outputs and acoustic context $Z_S$. Unlike Mask-CTC based approaches (Higuchi et al., 2020; 2021b), which operate on reduced intermediate CTC outputs and explicitly handle substitutions, insertions, and deletions, our method is more closely related to Nozaki & Komatsu (2021); Chi et al. (2021) and operates on un-reduced outputs, allowing implicit handling of all error types as well as leveraging full acoustic context, thereby relaxing the independence limitations in the correction phase. For a more detailed discussion in relation to the related works, please refer to appendix A.1.

The CTC logits $Y_{\text{CTC}}$, are detached from the computational graph, and the argmax is taken to produce the raw CTC predictions $\hat{Y}_{\text{CTC}}$. A subset of the tokens in $\hat{Y}_{\text{CTC}}$ is then masked to create the masked CTC outputs $\hat{Y}_{\text{CTC}}^M$. These masked CTC outputs are embedded into $Z_{\hat{Y}}^M$ and point-wise added with the unmasked speech embeddings $Z_S$, resulting in $X_{\text{corr}} = Z_{\hat{Y}}^M + Z_S$, where $X_{\text{corr}}$ represents the combined input for CTC error correction, which are passed through the shared Conformer backbone $\mathcal{F}$, and forwarded to the CTC correction head, $\mathcal{H}_{\text{CORR}}$, where the loss is computed using the CTC loss function:

$$Y_{\text{corr}} = \mathcal{H}_{\text{CORR}}(\mathcal{F}(X_{\text{corr}})), \tag{6}$$

$$\mathcal{L}_{\text{CTC-corr}} = -\log \sum_{\mathbf{a} \in \mathcal{A}(T)} P(\mathbf{a}|Y_{\text{corr}}). \tag{7}$$

By correcting the CTC predictions in this manner, we expect to mitigate the potential negative impact of multitask learning on ASR performance and improve transcription accuracy.

### 2.4.2 UNCONDITIONAL SPEECH MASKED LANGUAGE MODEL

Classifier-Free Guidance (CFG) (Ho & Salimans, 2021) has been used in diffusion models to balance conditional and unconditional outputs at each time step. This approach has also been successfully

applied to mask-predict models in image generation, such as Muse (Chang et al., 2023a). We adapt CFG to improve the TTS performance by incorporating an unconditional speech only MLM.

For speech MLM training, the masked speech embeddings $\boldsymbol{Z}_S^M$, are passed through the shared Conformer backbone, $\mathcal{F}$. The output is then used to predict the speech tokens at masked positions using a cross-entropy loss, similar to the text-to-speech task but without text conditioning and reusing the same speech head $\mathcal{H}_{\text{SPEECH}}$:

$$\boldsymbol{Y}_{\text{MLM}} = \mathcal{H}_{\text{SPEECH}}(\mathcal{F}(\boldsymbol{Z}_S^M)), \tag{8}$$

$$\mathcal{L}_{\text{MLM}} = -\sum_{i \in \mathcal{M}_{\text{speech}}} \boldsymbol{S}_i \log P(\boldsymbol{Y}_{\text{MLM},i}), \tag{9}$$

where $\mathcal{M}_{speech}$ is the set of masked positions $\boldsymbol{S}$ are the target speech tokens, and $P(\boldsymbol{Y}_{\text{MLM},i})$ is the predicted probability at position $i$. The goal during training is to reconstruct the masked speech tokens from the unmasked speech embeddings.

## 2.5 MASKING SCHEME

In our multitask framework, we employ different masking schemes depending on the task, as the requirements for masking vary. For the CTC error correction task, we use a uniform masking strategy, while for the TTS and speech MLM tasks, we utilize a more aggressive cosine-based masking strategy. Below, we explain the rationale for each scheme.

**CTC Error Correction Task: Uniform Masking**   During training for the CTC error correction task, the model refines low-confidence tokens without starting from fully masked tokens. A uniform masking strategy is applied, where the mask is sampled as:

$$\mathcal{M}_{\text{corr}} = \text{Bernoulli}(p), \quad p \sim \mathcal{U}(0,1), \tag{10}$$

targeting a random number of tokens from the CTC output $\hat{Y}_{\text{CTC}}$ for correction. This approach ensures the model focuses on low-confidence tokens while maintaining enough context for effective refinement.

**TTS and Speech MLM Tasks: Cosine Masking Schedule**   For the TTS and speech MLM tasks, we start with fully masked sequences and progressively unmask tokens using a cosine masking schedule inspired by MaskGIT (Chang et al., 2022). At each step, the mask is determined as:

$$\mathcal{M}_{\text{speech}} = \text{Bernoulli}(p), \quad p = \cos(u), \quad u \sim \mathcal{U}(0, \frac{\pi}{2}), \tag{11}$$

where $u$ is sampled from a uniform distribution over $[0, \frac{\pi}{2}]$, and $p$ is the cosine of $u$. This schedule mimics the inference iterative refinement where more tokens are masked initially and reduces masking as the model refines the outputs.

## 2.6 OVERALL TRAINING OBJECTIVE

In our framework, we adopted the simplest weighted-sum scalarization approach for multi-objective optimization, assigning equal weights to all task-specific losses from eqs. (2), (4), (5), (7) and (9) to obtain a single objective as follows:

$$\mathcal{L}_{\text{total}} = \mathcal{L}_{\text{CTC}} + \mathcal{L}_{\text{TTS}} + \mathcal{L}_{\text{MLM}} + \mathcal{L}_{\text{CTC-corr}} + \mathcal{L}_{\text{LEN}}, \tag{12}$$

This choice was based on the assumption that the tasks are cooperative rather than conflicting and the combined loss ensures that the model learns to balance multiple tasks efficiently, allowing it to leverage shared knowledge across tasks while optimizing performance for each individual task.

## 2.7 INFERENCE

### 2.7.1 SPEECH RECOGNITION

During inference, the speech embeddings $\boldsymbol{Z}_S$ are passed through the Conformer backbone into the CTC head to generate logits. Low-confidence tokens are identified based on a threshold $\tau$, masked and refined iteratively by combining masked outputs $\hat{Y}_{\text{CTC}}^M$ with the speech embeddings $\boldsymbol{Z}_S$. Refinement continues until all token confidence scores exceed $\tau$ or for a fixed number of iterations. The final transcription is obtained through CTC decoding of $\boldsymbol{Y}_{\text{corr}}$.

### 2.7.2 Text to Speech

For TTS, text tokens $\boldsymbol{T}$ are passed through the backbone into the length prediction head to predict log lengths, which are then converted to integer lengths as $\boldsymbol{L}_{\mathrm{pred}} = \lceil \exp(\boldsymbol{Y}_{\mathrm{LEN}}) \rceil$. These predicted lengths are used to upsample the text tokens, $\boldsymbol{T}_{\mathrm{up}} = \mathrm{Upsample}(\boldsymbol{T}, \boldsymbol{L}_{\mathrm{pred}})$, which are combined with fully masked speech embeddings $\boldsymbol{Z}_S^M$, and passed through the backbone into the speech prediction head. During generation, tokens are iteratively unmasked based on their confidence scores, starting with fully masked tokens. The unmasking follows the cosine schedule, where the masking probability at each iteration $t$ is given by $p_t = \cos\left(\frac{t\pi}{2T}\right)$.

Additionally, for Classifier-Free Guidance (CFG), we forward $\boldsymbol{Z}_S^M$ without text conditioning to obtain the unconditional logits $\boldsymbol{Y}_{\mathrm{MLM}}$. The final output logits are then a weighted combination of the conditional logits $\boldsymbol{Y}_{\mathrm{TTS}}$, and the unconditional logits $\boldsymbol{Y}_{\mathrm{MLM}}$, as follows:

$$\boldsymbol{Y}_{\mathrm{pred}} = (1 + \lambda) \cdot \boldsymbol{Y}_{\mathrm{TTS}} - \lambda \cdot \boldsymbol{Y}_{\mathrm{MLM}}, \tag{13}$$

where $\lambda$ is the guidance weight controlling the balance between conditional and unconditional outputs. This process yields the final speech sequence $\boldsymbol{S}_{\mathrm{pred}}$.

## 3 Experiments

To verify the effectiveness of our proposed methods, we perform extensive experimental evaluations. In this section, we first describe the training details followed by the various experimental results in the subsequent subsections.

### 3.1 Model Architecture and Training Details

#### 3.1.1 Training Infrastructure and Settings

All our models were implemented using the Pytorch(Paszke et al., 2019) framework and trained on 4 NVIDIA H100-80G GPUs with *bfloat16*(Burgess et al., 2019) precision. For efficient distributed training, we utilized *deepspeed*(Rajbhandari et al., 2020) ZeRO 2 optimization. For all experiments, we utilize the Adam optimizer(Kingma & Ba, 2015) with a peak learning rate of $2.5e^{-4}$, linearly warmed up over the first 4K iterations and decayed over the remaining iterations with a cosine schedule. The *beta1* and *beta2* of the optimizer are set to $\{0.8, 0.99\}$, with no weight decay, and clip the gradient norm with a maximum of 0.5.

#### 3.1.2 Training Datasets

For training all our models, we utilized the 60K hour LibriLight(Kahn et al., 2020) dataset. Additionally, for punctuated and cased transcriptions, we utilized the transcriptions provided in the LibriHeavy(Kang et al., 2024) dataset. We mainly use the 509 hour *small* subset of the LibriHeavy dataset for our experiments. We filtered out samples shorter than 0.2s and longer than 20s for better GPU utilization. Further, we removed samples which violate the input-target length constraints of the CTC loss. Following this we are left with around 500 hours of speech data that we use for the training. We perform all our experiments on 16Khz speech.

#### 3.1.3 Model Architecture

For all experiments (unless specified otherwise), we used the same conformer backbone with 6 layers with hidden size of 384, 8 attention heads, 1536 linear size, and convolution kernel size of 7. We utilize Rotary Position Embedding (RoPE) (Su et al., 2024). For each of the task-specific heads, we utilized the same structure of a linear layer with the same hidden size, followed by GELU activation and layer normalization, and finally an output linear layer to map to the respective output dimensions. We trained the models on the *small* subset of the LibriHeavy dataset for 30K iterations with a batch size of 64.

To convert the content tokens generated by our proposed method to acoustic tokens from the codec, we utilized the masked iterative generative model, SoundStorm(Borsos et al., 2023b). We reproduced this model by utilizing a conformer backbone with 12 layers with hidden size of 1024, 16

attention heads, 4096 linear size, and convolution kernel size of 5 with RoPE, followed by linear output heads for each RVQ level in the codec. We also trained this model on the LibriLight dataset with 15 second segments for 50K iterations with a batch size of 640.

### 3.1.4 TOKENIZERS

For the acoustic tokenizer, we use utilize a 12-level RVQ-based codec following the architecture of Descript Audio Codec (DAC) (Kumar et al., 2024), trained on the LibriLight dataset with a batch size of 144 for 200K iterations using the hyperparameters from[1]. For the content tokenizer, we utilize the publicly available HuBERT large checkpoint trained on the LibriLight dataset[2]. Further we trained a 1024-cluster K-Means tokenizer on the *train-clean-100* subset of the LibriSpeech(Panayotov et al., 2015) dataset. Both the acoustic and content tokenizers produce tokens at the rate of 50Hz, which simplifies the implementation. Finally for the text tokenization, we used a simple *utf-8* byte based representation. Byte representation has the advantage of having a compact output layer, requiring minimal text pre-processing and ability to handle any character and language. We did some basic pre-processing to expand certain common word contractions and numbers as is standard practice in ASR or TTS training.

### 3.2 EVALUATION DATASETS AND METRICS

To evaluate the performance of our proposed method, we utilized the following datasets and metrics. For evaluation of the ASR task, we utilize the LibriSpeech *test-clean* subset. For the TTS task, we randomly sampled 40 sentences from the LibriSpeech *test-clean* subset with at least 20 words in each sentence as the text input and sampled 3-8 second segments from all 20 speakers of the DAPS((Mysore, 2014) dataset as the speaker prompt for zero-shot TTS evaluation.

For evaluating the ASR performance, we utilized the Word Error Rate (WER) and Character Error Rate (CER). Since, our model is trained on punctuated and cased inputs, for the evaluation, we normalize the predicted sentences be removing all punctuation except the apostrophe and converting to upper-case following the format of the LirbiSpeech transcripts.

We utilized several automatic metrics for zero-shot TTS evaluation, which included UTMOS (Saeki et al., 2022) for speech quality, Speaker Encoder Cosine Similarity (SECS) using the *wavlm-base-plus-sv* model[3] for speaker similarity, Character Error Rate (CER) using the *hubert-large-ls960-ft* model[4] for percieved intelligibility and robustness. To effectively evaluate the zero-shot TTS performance, we finally perform a subjective evaluation. For this evaluation, we utilize one sample per speaker from the evaluation data described above, and present samples from two systems along with the text input and ask the raters to score between $\{+2, -2\}$ (order of methods is randomized), based on *naturalness, acoustic quality, and human likeness*. At least 5 raters rate each sample and this gives us the Comparative mean opinion score (CMOS). Similarly we perform subjective evaluation for Speaker similarity Comparative mean opinion score (SCMOS), the difference being for this evaluation the reference speaker prompt is also presented to the raters and asked to compare which of the two shown methods is more similar to the reference. The rating scale and number of raters per sample are same as the CMOS evaluation. For all scores, we compute the $95\%$ confidence interval by bootstrapping and measure the statistical significance with the Wilcoxon signed-rank test (p-value).

To evaluate latency, we conducted controlled inference runtime (IR) experiments. For TTS, we measured end-to-end runtime (IR-e2e) and text-to-content token runtime (IR-t2c) using a 405-character sentence. For ASR, we measured IR with a 60-second sample. All measurements were averaged over 100 trials on a single H100 GPU.

### 3.3 BASELINES

To test the effectiveness of our method, we utilize several state-of-the-art baselines across various speech synthesis and recognition paradigms.

---

[1]https://github.com/descriptinc/descript-audio-codec/blob/main/conf/final/16khz.yml

[2]https://huggingface.co/facebook/hubert-large-ll60k

[3]https://huggingface.co/microsoft/wavlm-base-plus-sv

[4]https://huggingface.co/facebook/hubert-large-ls960-ft

1. Speech Synthesis
   (a) Non-iterative: HierSpeech++(Lee et al., 2023), YourTTS(Casanova et al., 2022)
   (b) Diffusion: StyleTTS2 (Li et al., 2024)
   (c) Autoregressive: XTTS(Casanova et al., 2024), WhisperSpeech[5] (based on (Borsos et al., 2023a; Kharitonov et al., 2023))
2. Speech Recognition
   (a) Transducer based non-discrete ASR: Zipformer-Transducer(Yao et al., 2023)[6]
   (b) Discrete ASR: Coformer-CTC trained on discrete content tokens.

We utilize publicly released checkpoints of all the above methods except SoundStorm, which we reproduce and train on the same dataset as our method and with similar parameter count.

## 3.4 ZERO-SHOT TEXT TO SPEECH SYNTHESIS

### 3.4.1 ABLATION STUDY

Table 1: Zero-shot TTS ablation study for different tasks.

| Task Setting | UTMOS | CER | SECS |
|---|---|---|---|
| w SMLM, w CORR | $4.39 \pm 0.04$ | 0.95 | $0.94 \pm 0.01$ |
| w SMLM, w/o CORR | $4.41 \pm 0.04$ | 1.08 | $0.94 \pm 0.01$ |
| w/o SMLM, w/o CORR | $4.39 \pm 0.03$ | 0.82 | $0.94 \pm 0.01$ |

To study the effect of each component of our proposed method for the TTS task, we conduct extensive ablation study. First we check the performance of introducing the auxiliary CTC-correction and Speech MLM tasks. For this evaluation, we predict the content tokens in a single pass through the network. The results in Table 1 indicate that the auxiliary tasks have minimal impact on the core TTS task and the performance is maintained.

Table 2: Zero-shot TTS ablation study for different number of iterations.

| Iters | UTMOS | CER | SECS |
|---|---|---|---|
| 1 | $4.39 \pm 0.04$ | **0.95** | $0.94 \pm 0.01$ |
| 4 | **$4.43 \pm 0.03$** | 1.12 | $0.94 \pm 0.01$ |
| 8 | $4.41 \pm 0.03$ | 1.23 | $0.94 \pm 0.01$ |

Table 3: TTS ablation study for CFG weight $\lambda$.

| $\lambda$ | UTMOS | CER | SECS |
|---|---|---|---|
| 0.0 | **$4.43 \pm 0.03$** | 1.12 | $0.94 \pm 0.01$ |
| 1.0 | **$4.43 \pm 0.02$** | **0.55** | $0.94 \pm 0.01$ |
| 1.5 | $4.40 \pm 0.04$ | 0.95 | $0.94 \pm 0.01$ |
| 2.0 | $4.42 \pm 0.02$ | 0.69 | $0.94 \pm 0.01$ |

Next we evaluate the number of iterations for generating the content tokens. For this task we do not use CFG. The results reported in Table 2 show that our method is able to achieve very good performance even with a single step. Increasing the number of iterations slightly improves the UTMOS, while slightly degrading the CER. For subsequent experiments, we utilize 4 iterations, which provides a balance between the UTMOS and CER scores.

Finally we evaluate the effect of CFG weight, $\lambda$. From the results in Table 3, we can see that using CFG significantly improves the CER while maintaining the speech quality with $\lambda = 1.0$ giving the best CER score. The degradation observed in single-pass inference with SMLM is marginal (Table 1), and is compensated by the iterative inference with CFG, which fully utilizes SMLM's benefits, significantly improving CER for TTS.

### 3.4.2 MAIN RESULT

Finally, with the best settings from the ablation study, we compare our method with state-of-the-art methods in Table 4. Our method significantly outperforms baselines trained on similar-scale data.

---

[5]https://github.com/collabora/WhisperSpeech
[6]https://github.com/k2-fsa/icefall/tree/master/egs/libriheavy/ASR

Table 4: Zero-shot TTS performance comparison. Methods with * indicate multilingual models. UD refers to Unpaired Data while PD refers to Paired Data in hours.

| | UD | PD | UTMOS | CER | SECS | IR-e2e (s) | IR-t2c (s) |
|---|---|---|---|---|---|---|---|
| *Large scale paired data* | | | | | | | |
| HierSpeech++* | 500k | 2.8k | **4.46 ± 0.02** | 0.88 | **0.94 ± 0.01** | 0.16 ± 0.00 | - |
| XTTS* | - | 27k | 4.12 ± 0.07 | 0.78 | 0.93 ± 0.01 | 2.60 ± 0.03 | - |
| WhisperSpeech | 60k | 60k | 3.95 ± 0.11 | 0.66 | 0.93 ± 0.01 | 17.91 ± 0.04 | 2.84 ± 0.01 |
| *Small scale paired data* | | | | | | | |
| YourTTS* | - | 689 | 3.69 ± 0.08 | 2.02 | 0.90 ± 0.02 | **0.11 ± 0.00** | - |
| StyleTTS2 | 94k | 245 | 4.43 ± 0.03 | 1.59 | 0.91 ± 0.02 | 0.27 ± 0.00 | - |
| Ours | 60k | 500 | 4.43 ± 0.02 | **0.55** | **0.94 ± 0.01** | 0.57 ± 0.00 | **0.06 ± 0.00** |

Additionally, it surpasses systems trained on thousands of hours of paired speech, demonstrating superior robustness in terms of the CER score and data efficiency. A point to note here is that the UTMOS and SECS metrics are more influenced by the SoundStorm and Codec rather than our method. IR-t2c shows that T2V2 is $\sim$ 5 orders of magnitude faster than the AR WhisperSpeech baseline and surpasses other AR methods in IR-e2e.

We present the result of the subjective evaluation in Table 5. From the results we can see that in terms of CMOS, our method is at par with all the state-of-the-art methods since the p-value is greater than 0.05 indicating no statistically significant difference. While for the SCMOS, our method outperforms XTTS and WhisperSpeech while being at par with HierSpeech++ and StyleTTS2. We encourage the readers to listen to the samples provided in the supplementary materials.

Table 5: Comparative MOS for Speech Quality (CMOS) and Speaker Similarity (SCMOS) on a scale $\{-2, +2\}$. p-value $\leq 0.05$ indicate statistical significance.

| | CMOS (p-value) | SCMOS (p-value) |
|---|---|---|
| HierSpeech++ | **+0.10 ± 0.25 (0.337)** | **+0.12 ± 0.26 (0.287)** |
| XTTS | **-0.13 ± 0.28 (0.418)** | -0.30 ± 0.22 (0.007) |
| StyleTTS2 | **+0.16 ± 0.25 (0.271)** | **+0.14 ± 0.24 (0.201)** |
| WhisperSpeech | **-0.11 ± 0.27 (0.490)** | -0.63 ± 0.21 ($1.5e^{-7}$) |
| Ours | **0.00** | **0.00** |

### 3.5 AUTOMATIC SPEECH RECOGNITION

#### 3.5.1 ABLATION STUDY

To evaluate the impact of various components and inference settings on ASR performance, we conducted an ablation study. In Table 6, we examine the effect of auxiliary tasks on ASR performance, noting that the CTC Correction task notably regulates the model and improves raw ASR performance from the CTC head without correction. While the SMLM task alone slightly degrades performance, the CORR task compensates, allowing iterative correction to significantly enhance ASR results.

Next, we investigated the confidence threshold and number of iterations for CTC error correction (Table 7). A threshold of 0.7 with 16 correction iterations yielded the best results. Increasing the iterations slightly raised the IR, but the process remained efficient.

Finally, we analyzed how iterative correction handles all error types (Table 8). Using the best configuration from Table 7, we observed consistent relative improvements across substitutions, insertions, and deletions, demonstrating the effectiveness of our proposed mechanism.

#### 3.5.2 MAIN RESULT

Finally we compare our method with the baselines in Table 9. We show the results of the non-discrete reference model Zipformer-Transducer also trained with casing and punctuation. As we can

Table 6: ASR ablation study for different tasks.

|  | CER | WER |
|---|---|---|
| w SMLM, w CORR | **2.732** | **8.651** |
| w SMLM, w/o CORR | 2.949 | 9.428 |
| w/o SMLM, w/o CORR | 2.886 | 9.120 |

Table 7: ASR ablation study for different correction thresholds and iterations.

| Corr. Thresh | Iters | CER | WER | IR(s) |
|---|---|---|---|---|
| w/o CORR | - | 2.73 | 8.65 | 0.32 ± 0.02 |
| 0.8 | 1 | 2.73 | 8.44 | 0.40 ± 0.03 |
| 0.8 | 4 | 2.72 | 8.37 | 0.39 ± 0.03 |
| 0.8 | 8 | 2.72 | 8.33 | 0.42 ± 0.03 |
| 0.7 | 8 | 2.72 | 8.29 | 0.42 ± 0.03 |
| 0.7 | 16 | **2.71** | **8.27** | 0.47 ± 0.03 |
| 0.7 | 32 | **2.71** | **8.27** | 0.60 ± 0.03 |

Table 8: Individual error type improvements.

|  | Sub | Ins | Del |
|---|---|---|---|
| w/o CORR | 1.300 | 0.090 | 0.140 |
| w CORR | **1.255** ↓3.46% | **0.082** ↓8.89% | **0.135** ↓3.57% |

see discrete ASR still under-performs compared to non-discrete AR zipformer-transducer, however when compared with the discrete baseline (Conformer-CTC), we achieve comparable performance. We further verified how our method scales with scaling the data. For that we trained a larger version of our model from scratch with 10 layers on the LibriHeavy large subset (50K hours) and we see that our method outperforms the CTC baseline with same 10-layer conformer backbone. Thus validating the effectiveness and scaling capabilities of T2V2.

Table 9: ASR results for models trained with punctuation and casing. The publicly released models for Zipformer-Transducer are used for the evaluation, while Conformer-CTC is trained by us.

|  | Libriheavy Subset | CER | WER | IR (s) |
|---|---|---|---|---|
| *Non-discrete ASR (BPE encoding)* |  |  |  |  |
| Zipformer-Transducer | small | 2.01 | 5.33 | 1.49 ± 0.07 |
| Zipformer-Transducer | large | 0.66 | 1.99 | 1.51 ± 0.14 |
| *Discrete ASR (Byte Encoding)* |  |  |  |  |
| Conformer-CTC | small | **2.69** | 8.28 | 0.32 ± 0.02 |
| Ours | small | 2.71 | **8.27** | 0.47 ± 0.03 |
| Conformer-CTC | large | 1.53 | 4.36 | 0.34 ± 0.02 |
| Ours | large | **1.31** | **4.09** | 0.55 ± 0.02 |

## 4 FUTURE PERSPECTIVES AND LIMITATIONS

In the current work, we focus on T2V2's capability in the two core tasks of ASR and TTS, however, the multitask training paradigm could potentially allow the model to learn high performing and general purpose representations of speech and text which could be used for other downstream tasks as well as enable training modality agnostic language models. As for the limitations of the T2V2, we see that discrete ASR still underperforms compared to models trained with continuous features, while for TTS, we rely on separate content token to acoustic token translation, which might limit the quality of synthesized speech.

## 5 CONCLUSIONS

In this work, we introduced T2V2, the first unified non-autoregressive model capable of handling both automatic speech recognition and text-to-speech synthesis. By leveraging discrete tokens from self-supervised models and incorporating auxiliary tasks like CTC error correction and unconditional speech MLM, T2V2 effectively improves performance across both ASR and TTS. Additionally, the use of Monotonic Alignment Search ensures a self-contained approach to text-speech alignment without relying on external aligners. Our experimental results demonstrate significant gains in TTS quality while maintaining competitive ASR performance, highlighting the potential of unified models for multitask learning in speech processing.

## ETHICS STATEMENT

Our proposed method achieves high quality synthetic speech in a target speaker's voice with a short sample, thus it is paramount to address the broader impact of our research. Our method may be potentially misused by malicious parties, towards this end we verified that our speech could still be detected as fake speech by a third party detector[7].

## REPRODUCIBILITY STATEMENT

For reproducibility, we have carefully described the model architecture, optimization tasks, training masking schemes, loss functions, and inference procedure in Section 2. Additionaly, implementation and training hyperparameters, and all pre-trained models have been described in Section 3.1, including their links in the footnotes.

## ACKNOWLEDGMENTS

This work was partially supported by JST Moonshot R&D Grant Number JPMJPS2011, CREST Grant Number JPMJCR2015 and Basic Research Grant (Super AI) of Institute for AI and Beyond of the University of Tokyo.

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

## A APPENDIX

### A.1 CTC ERROR CORRECTION AND RELATED WORK

#### A.1.1 RELATION TO ALIGN-REFINE

Align-Refine (Chi et al., 2021) and our method share the conceptual similarity of utilizing an iterative refinement paradigm. However, our approach differs significantly in its formulation:

**Separate Decoder vs. Direct Approach:** Align-Refine employs a separate non-causal transformer decoder for refinement, integrating encoder features via cross-attention. In contrast, our method formulates refinement as a separate task by reusing the same backbone encoder and directly combining intermediate CTC outputs with speech tokens for refinement. This eliminates the need for additional modules such as decoders or cross-attention layers.

#### A.1.2 RELATION TO SELF-CONDITIONED CTC

Our method also shares similarities with Self-Conditioned CTC (Nozaki & Komatsu, 2021) in leveraging intermediate CTC outputs, but there are notable differences:

**Intermediate Conditioning:** Self-Conditioned CTC applies intermediate CTC logit embeddings via additional linear projection layers at intermediate encoder stages, making it well-suited for dedicated ASR models. However, in our multi-task scenario, where the backbone supports both ASR and TTS tasks, such intermediate conditioning might not generalize well. Our approach avoids additional layers, maintaining a lightweight design.

**Iterative Refinement:** Unlike Self-Conditioned CTC, which does not support iterative refinement, our method incorporates iterative refinement as a core mechanism.

