# OpenReview forum: "T2V2: A Unified Non-Autoregressive Model for Speech Recognition and Synthesis via Multitask Learning"
_ICLR.cc/2025/Conference — ICLR 2025 Poster_

### Official Review · Reviewer_AXxb · 2024-10-28

**Soundness:** 2
**Presentation:** 3
**Contribution:** 2
**Rating:** 6
**Confidence:** 4

**Summary:**

This paper proposes a unified non-autoregressive model to perform both automatic speech recognition (ASR) and text-to-speech (TTS) synthesis tasks. CTC error correction and classifier free are used to improve ASR and TTS performance respectively. Intermediate CTC outputs are used to align text and speech by Monotonic Alignment Search, without relying on external aligners.

**Strengths:**

1. It is interesting to use the intermediate CTC output to align text and speech via monotonic alignment search without relying on an external aligner.
2. The CTC error correction method is useful and may be applied to other state-of-the-art ASR systems.
3. The paper is well-structured and written.

**Weaknesses:**

1. Figure 1 is too complicated, with the arrows flowing in interlocking directions, so that readers may feel difficult to follow it.
2. The content tokens employed in the T2V2 model are derived from HuBERT. However, converting these tokens into acoustic tokens and subsequently into speech requires additional components, namely a semantic-to-speech model (specifically SoundStorm) and a codec decoder. In my view, the efficacy of the speech synthesis process is largely dependent upon the performance of SoundStorm.
3. The models being compared in the TTS experiment should have models that only predict content tokens + SoundStorm, which is the most similar setting to T2V2 + SoundStorm.
4. In Tables 1 and 6, when comparing the performance of (with SMLM, without COR) and (without SMLM, without COR), it appears that SMLM harms both TTS and ASR tasks.
5. The change in Corr. thresholds and iterations in Table 7 doesn't seem to make the CER change much.

**Questions:**

1. What is the time increase for 16 iterations compared to 1 iteration in Table 7?
2. What is the performance of the large model for TTS tasks?
3. Table 2 seems to have the best overall performance with iter=1.

---

> ### Author Response · Authors · 2024-11-19
> **Official Comment by Authors (1/2)**
>
> **We thank the reviewer for the detailed feedback and constructive questions. We are encouraged by the recognition of the strengths of our method and clarity of writing.**
> Below we first list the revision plan based on the queries and suggestions of the reviewer, followed by detailed responses to each query.
>
> ---
>
> **Revision Plan**
> (Please check detailed responses for more information):
> 1. Update and simplify Fig 1.
> 2. Add inference runtime comparisons (Tables 4, 7, and 8).
>
> **NOTE**: The changes are colored in **blue** text in the revised manuscript for easy visibility.
>
> ---
>
> **Weaknesses**
>
> **Figure 1 is too complicated, with the arrows flowing in interlocking directions, so that readers may feel difficult to follow it.**
> Thank you for pointing this out, we will revise the figure to make it simpler to understand.
>
> ---
>
> **The efficacy of the speech synthesis process is largely dependent upon the performance of SoundStorm.**
> The performance of the T2V2 model depends partly on the SoundStorm model for content-to-acoustic token conversion. SoundStorm was chosen for its efficiency as an iterative NAR method, which aligns well with our framework. However, we acknowledge that alternative models, such as Grouped Masked Modeling (predicting grouped acoustic levels) **[1]** or PolySpeech detokenizer (using a VITS-like structure with adversarial loss) **[2]**, could potentially offer further improvements. Exploring these alternatives is part of our future work.
>
> Moreover, we have included inference runtime comparisons with the baselines (Tab. 4), especially comparing only the text-to-content tokens stage of T2V2 with that of WhisperSpeech, which employs AR modeling for this stage. We can see that our method is significantly faster.
>
> Please also refer to the “Inference Runtimes” comment at the top.
>
> ---
>
> **The models being compared in the TTS experiment should have models that only predict content tokens + SoundStorm, which is the most similar setting to T2V2 + SoundStorm.**
> The models we compared in the TTS experiments include dual-stage methods like Hierspeech++ and WhisperSpeech, which also predict content features before acoustic features. While WhisperSpeech uses autoregressive modeling, it shares the dual-stage tokenized modeling paradigm with our approach. Additionally, it is trained on the same dataset (LibriLight), making the comparison relevant. The other baselines are the current state-of-the-art methods, thus demonstrating the competitiveness of our method.
>
> ---
>
> **In Tables 1 and 6, when comparing the performance of (with SMLM, without COR) and (without SMLM, without COR), it appears that SMLM harms both TTS and ASR tasks.**
> The slight degradation observed in single-pass inference with SMLM is marginal, as iterative refinement (not used in single-pass inference) fully utilizes SMLM’s benefits. During iterative inference, CFG (enabled by SMLM) significantly improves TTS performance, and the iterative CTC correction (enabled by the CORR task) compensates for the slight single-pass degradation. Thus, emphasizing that SMLM is a crucial component of our proposed framework.
>
> ---
>
> **The change in Corr. thresholds and iterations in Table 7 doesn't seem to make the CER change much.**
> While average CER changes are minimal, small character corrections can significantly affect WER, as correct words (coarser units) lead to greater overall improvements as seen in the improved WER.
>
> ---
>
> Continued in the next comment ...

---

> > ### Author Response · Authors · 2024-11-19
> > **Official Comment by Authors (2/2)**
> >
> > **Questions**
> >
> > **What is the time increase for 16 iterations compared to 1 iteration in Table 7?**
> > We will provide an analysis of the time increase for 16 iterations compared to 1 iteration to demonstrate the computational trade-offs in Tables 4, 7, and 8 in the updated manuscript.
> >
> > Please also refer to the “Inference Runtimes” comment at the top.
> >
> > ---
> >
> > **What is the performance of the large model for TTS tasks?**
> >
> > | UTMOS         | CER   | SECS  |
> > |---------------|-------|-------|
> > | 4.41 ± 0.02   | 0.63  | 0.94  |
> >
> > We observed that while with a larger dataset and longer training, the ASR performance improves significantly, the TTS performance is saturated around the same level as the small model. This could be attributed to the fact that larger training datasets help the CTC model to recognize rare words better; however, for TTS, since it works with aligned text tokens, the impact of rare words is not severe as the model can learn to handle the sequence of expanded characters even with smaller training sets.
> >
> > ---
> >
> > **Table 2 seems to have the best overall performance with iter=1.**
> > In Table 2, CFG is not applied, and thus iterative inference does not leverage SMLM’s benefits. In Table 3, the iterative inference results demonstrate the gains achieved with CFG enabled by SMLM. Separating the ablations emphasizes the importance of SMLM and the strength of CFG in our framework.
> >
> > ---
> >
> > **Thank you for recognizing the strengths of our work and providing valuable feedback. We will incorporate these revisions to enhance the clarity, rigor, and completeness of the manuscript.**
> >
> > ---
> >
> > **References**
> > 1. Jeong, Myeonghun, et al. "Efficient Parallel Audio Generation Using Group Masked Language Modeling." IEEE Signal Processing Letters (2024).
> > 2. Yang, Runyan, et al. "PolySpeech: Exploring Unified Multitask Speech Models for Competitiveness with Single-task Models." arXiv preprint arXiv:2406.07801 (2024).

---

> > > ### Comment · Reviewer_AXxb · 2024-11-22
> > >
> > > I really appreciate the efforts of the authors, including revising Figures, and explaining the benefits from SMLM. If the authors can integrating these explanations (SMLM & CFG) directly into Section 3 which may increase the readability of paper, i will increase my score.
> > > However, I still think the results in Table 7 are not much different, even for WER, it's a difference of 0.11 and 0.02.

---

> > > > ### Author Response · Authors · 2024-11-22
> > > > **Response to Reviewer Comment**
> > > >
> > > > Thank you for your valuable feedback and thoughtful suggestions. We appreciate your recognition of our efforts in revising the figures and explaining the benefits of SMLM and CFG. Following your suggestion, we have integrated explanations of the benefits of SMLM and CFG directly into the respective sections of the paper to improve readability and clarity. Specifically:
> > > > - The single-pass degradation and the significance of the SMLM task for TTS have been addressed in **Section 3.4.1 (line 423)**.
> > > > - The impact and clarification of SMLM for ASR have been addressed in **Section 3.5.1 (line 476)**.
> > > >
> > > > We believe these additions provide a clearer understanding of the contributions and practical significance of these components.
> > > >
> > > > Regarding the results in Table 7, we agree that the differences between the various thresholds within the iterative correction method are not very different. However, we would like to emphasize that, compared to the result without applying correction (first row in Table 7), the iterative correction improves the WER by approximately 0.4. Additionally, the CORR task itself improves the WER by approximately 0.5 (as shown when comparing rows 1 and 3 in Table 6). Together, they yield a cumulative WER improvement of approximately 0.9. We believe this represents a significant enhancement.
> > > >
> > > > In future work, we aim to further enhance ASR accuracy and effectiveness by building on these results and exploring new avenues for improvement. Once again, we are grateful for your constructive comments, which have guided us in making our work clearer and more impactful.

---

> > > > > ### Comment · Reviewer_AXxb · 2024-11-22
> > > > >
> > > > > thanks for your further explanation. I have raised my score.

---

> > > > > > ### Author Response · Authors · 2024-11-25
> > > > > > **Thank You**
> > > > > >
> > > > > > Dear Reviewer AXxb,
> > > > > >
> > > > > > We sincerely thank you for your thoughtful feedback and for taking the time to review our rebuttal and update your evaluation. Your insights and suggestions have been invaluable in helping us improve our manuscript. We deeply appreciate your engagement and the effort you’ve put into reviewing our work.
> > > > > >
> > > > > > Best regards

---

### Official Review · Reviewer_D6js · 2024-11-04

**Soundness:** 2
**Presentation:** 2
**Contribution:** 2
**Rating:** 5
**Confidence:** 5

**Summary:**

This paper proposed a T2V2 framework that uses a shared Conformer backbone to jointly train non-autoregressive ASR and TTS tasks with Connectionist Temporal Classification (CTC) and masked language modeling (MLM) loss. Two axillary losses are additionally introduced to improve the model performance, which are CTC error correction to refine raw ASR outputs using contextual information from speech embeddings, and unconditional speech MLM, enabling classifier free guidance. In the proposed framework, Monotonic Alignment Search (MAS) was used with intermediate CTC outputs for TTS alignment without relying on external tools. The experiments show that T2V2 markedly improves synthesized speech robustness achieving state-of-the-art TTS performance and competitive discrete ASR performance.

**Strengths:**

The strengths of this paper are:

1. The first paper to unify ASR and TTS in one model that operates on discrete tokens.
2. Extensive experiments to show the performance of the proposed model.

**Weaknesses:**

The weaknesses of the paper are:

1. Need improvement in clarity.
2. Lacking validations on metrics like RTF or latency.
3. The performance does not seem to be competitive.

See below for details.

**Questions:**

The idea of unifying ASR and TTS in one model is interesting. However, my personal feeling is that the work is still in an early stage and is not ready for publication in ICLR as a top conference. Here are my concerns on the paper.

1. The clarity of the paper needs improvement.
- The paper involved many topics in the field of ASR and TTS such as discrete tokens, vocoder, non-autoregressive generation, etc. I understand it is hard to clarify each topic in detail. I'd suggest the authors to at least detail the used method for the claimed contributions. For example, MAS is claimed to be a major contribution. However, it is not clear how MAS is applied to the framework. How does it compare to the external alignment? In addition, I am wondering how it is compared to CTC alignment, which also provides speech-text alignments for non-autoregressive ASR (R. Fan, W. Chu, P. Chang and J. Xiao, "CASS-NAT: CTC Alignment-Based Single Step Non-Autoregressive Transformer for Speech Recognition," ICASSP 2021 - 2021 IEEE International Conference on Acoustics, Speech and Signal Processing (ICASSP), Toronto, ON, Canada, 2021, pp. 5889-5893, doi: 10.1109/ICASSP39728.2021.9413429.).
- It is suggested to clarify the loss for each head in Figure 1, especially the conditional TTS and unconditional MLM. Please differentiate the input and output, maybe with different colors?
- The experiments of hyperparameter tuning can be presented in the appendix.

2. Validation and performance
- A very important issue of the paper is the comparison of latency. One of the advantages of using both non-autoregressive generation and discrete token for ASR is the inference speed. The authors have no such comparison. In addition, the paper seems to use iterative generation during inference, the model should be rigorously classified as semi-autoregressive model instead of non-autoregressive model.
- The CTC error correction does not seem to be novel except that it is multi-tasked training with TTS loss. The Lcorr is not in-depth explored in this study. For example, according to the description of the paper, only the substation errors in CTC logits are considered to be corrected, while deletion and insertion errors are considered in literature. (Y. Higuchi, H. Inaguma, S. Watanabe, T. Ogawa and T. Kobayashi, "Improved Mask-CTC for Non-Autoregressive End-to-End ASR," ICASSP 2021 - 2021 IEEE International Conference on Acoustics, Speech and Signal Processing (ICASSP), Toronto, ON, Canada, 2021, pp. 8363-8367, doi: 10.1109/ICASSP39728.2021.9414198.)
- For TTS, the ablation shows that the impacts of MLM and CORR are low. It is interesting to see the performance of training the model with TTS only. This would provide us insights whether the joint ASR and TTS modeling is beneficial.
- For ASR, the performance on test-clean doesn't seem to be competitive, compared to non-discrete ASR (almost twice of the WER). The advantage of discrete ASR is not stated in the paper.
- What is the motivation to use CER for the English ASR evaluation?

Overall, the paper has large room for improvements.

---

> ### Author Response · Authors · 2024-11-19
> **Official Comment by Authors (1/3)**
>
> **We thank the reviewer for the detailed feedback and constructive questions.**
> Below we first list the revision plan based on the queries and suggestions of the reviewer, followed by detailed responses to each query.
>
> ---
>
> **Revision Plan**
> (Please check detailed responses for more information):
> 1. Include details of how MAS is applied in our framework (Sec. 2.3.2).
> 2. Update and simplify Fig. 1.
> 3. Add inference runtime comparisons (Tables 4, 7, and 8).
> 4. Include a discussion positioning the novelty of our CTC correction formulation in comparison to Mask-CTC and Improved Mask-CTC (Sec. 2.4.1).
> 5. Include a discussion on the advantages of discrete ASR (Sec. 1).
>
> **NOTE**: The changes are colored in **blue** text in the revised manuscript for easy visibility.
>
> ---
>
> **Weaknesses and Questions**
>
> **Clarity**
>
> **It is not clear how MAS is applied to the framework.**
> We acknowledge that the description of how we applied MAS to our framework is inadequate. We have updated the manuscript with a more detailed description of it, as well as updated the figure to reflect the MAS.
>
> We consider the CTC log probabilities as the alignment score by gathering the log probability corresponding to each text token in the text sequence for each time step in the intermediate CTC output. This gives us the alignment score matrix, which is then passed to the original MAS algorithm to find the best monotonic alignment.
>
> ---
>
> **How does it compare to the external alignment?**
> While it is possible to use external alignments, the external alignments are computed from the features of a separate model, which might not exactly align with the discrete representations that we work on. Unifying the ASR and TTS tasks provides us with alignments that are more consistent with the input representation since both tasks share the same representation.
>
> ---
>
> **How it is compared to CTC alignment, which also provides speech-text alignments for non-autoregressive ASR?**
> Thank you for bringing the CASS-NAT **[1]** paper to our attention. Upon reading the paper, here are our observations:
>
> Both the Viterbi-alignment used in CASS-NAT and our MAS with intermediate CTC log probabilities would result in obtaining similar alignments since even though the monotonicity is not directly enforced in the Viterbi alignment, the intermediate outputs are being trained with CTC loss which ensures monotonicity.
>
> The key difference is in the use of these alignments. We utilize a straightforward method of upsampling the text tokens by repetition for TTS task input, while CASS-NAT uses it to effectively reduce the sequence length for the  ASR decoder input.
>
> While the purposes of using the alignments are different, we believe a discussion about the CASS-NAT’s application of alignment and our CTC correction task is interesting and warranted (even though we do not use alignments for the correction task). CASS-NAT’s application of the alignment can be seen as a means of length-adaptive soft-copy of encoder features for the decoder input (since, in their framework, the NAR decoder cannot be trained with teacher forcing) to further enhance the intermediate CTC outputs. However, since the decoder input is dependent on the predicted reduced sequence length, CASS-NAT requires special tricks like Error-based sampling (Sec. 2.6) during inference to give the decoder some flexibility. In contrast, our CTC correction mechanism has more room to maneuver since we perform the iterative masked correction on un-reduced intermediate CTC outputs, essentially enabling the handling of insertion and deletion errors implicitly in addition to substitution errors.
>
> ---
>
> **It is suggested to clarify the loss for each head in Figure 1.**
> Thank you for pointing this out, we will revise the figure to make it simpler to understand and all inputs and outputs clearly demarked.
>
> ---
>
> **The experiments of hyperparameter tuning can be presented in the appendix.**
> We acknowledge the suggestion to include hyperparameter tuning details in the appendix. However, we did not perform extensive tuning due to the significant computational costs involved. The main tuning efforts focused on the auxiliary tasks, and these results have already been presented in the ablation studies.
>
> For other hyperparameters, such as learning rate and batch size, we relied on fairly standard values commonly used in the literature and did not conduct additional tuning experiments. We will clarify this in the revised manuscript.
>
> ---
>
> Continued in the next comment ...

---

> > ### Author Response · Authors · 2024-11-19
> > **Official Comment by Authors (2/3)**
> >
> > **Validation and Performance**
> >
> > **A very important issue of the paper is the comparison of latency.**
> > We acknowledge the need for inference time comparisons with AR models, and we did not originally include these since it is well-known that NAR models are faster than AR models for inference. However, we now realize that it is important to demonstrate the inference efficiency our approach explicitly, and will include inference runtime measurements in the revised manuscript.
> >
> > Please also refer to the “Inference Runtimes” comment at the top.
> >
> > ---
> >
> > **The model should be rigorously classified as a semi-autoregressive model instead of a non-autoregressive model.**
> > Thank you for raising this point regarding the classification of our model. While we do employ iterative refinement during inference, we believe that our approach is better classified as an iterative non-autoregressive (NAR) model rather than semi-autoregressive.
> >
> > The distinction lies in the absence of causality constraints in our method. Iterative-NAR models, like ours, allow full access to both unmasked past and future contexts during decoding, unlike semi-autoregressive models, which impose autoregressive constraints within sub-sequences.
> >
> > ---
> >
> > **The CTC error correction does not seem to be novel except that it is multi-tasked training with TTS loss.**
> > While the concept of CTC error correction is not novel, we believe our formulation provides unique advantages. Existing approaches like Mask-CTC and Improved-Mask-CTC operate on reduced intermediate CTC outputs, explicitly handling substitutions, insertions, and deletions.
> >
> > Our method, in contrast, works on un-reduced intermediate outputs, allowing implicit handling of all three error types. Additionally, our decoder integrates acoustic context (content token embeddings) alongside partial text predictions, enhancing correction accuracy. We will update Sec. 2.4.1 to reflect these distinctions.
> >
> > ---
> >
> > **For TTS, the ablation shows that the impacts of MLM and CORR are low. It is interesting to see the performance of training the model with TTS only.**
> > As we do not use external alignments, the TTS task relies on CTC alignment during training, making TTS-only training infeasible in our framework.
> >
> > Regarding the impacts of SMLM and CORR tasks on TTS in the ablation study (Tab. 1), it shows that the auxiliary tasks do not harm the performance of the core TTS task. The results in this table are computed with a single pass. However, Table 3 shows the advantage of combining the iterative generation with CFG (enabled by the SMLM task) to significantly improve the CER of synthesized speech.
> >
> > ---
> >
> > **For ASR, the performance on test-clean doesn't seem to be competitive, compared to non-discrete ASR.**
> > We acknowledge that discrete ASR performance on test-clean is not as competitive as continuous ASR. Moreover, the continuous baseline is an AR transducer-based model, which is well known to perform better than CTC. The combined effects of using continuous features and AR decoding with a dedicated ASR transducer model result in much better performance. Thus, it is not an entirely fair comparison; we chose these baselines since they were the only publicly available models trained with continuous features and with punctuation and casing on the same Libriheavy subsets as ours. Even though our results are worse than the continuous transducer model, according to [2], WER below 10% is usable and our method provides a balanced tradeoff between inference speed and accuracy (please see inference runtimes in the updated paper).
> >
> > For a more detailed analysis of the comparison of discrete and non-discrete ASR, we refer to the related work [3], where they apply several tricks to close the gap between the two. In the current work, our focus is primarily on unifying the two speech-related tasks of ASR and TTS rather than getting absolute SOTA performance.
> >
> > ---
> >
> > **The advantage of discrete ASR is not stated in the paper.**
> > Discrete ASR offers several advantages, as highlighted in related work [2]. It enables significantly smaller storage and transmission sizes compared to raw waveforms, making it particularly useful for handling large datasets. Additionally, discrete tokens provide an intermediate representation that captures both acoustic and linguistic information while being less speaker-specific, making it well-suited for the unified modality required in our framework. We will update the manuscript to explicitly state these advantages.
> >
> > ---
> >
> > **What is the motivation to use CER for the English ASR evaluation?**
> > CER highlights improvements in fine-grained error correction during intermediate CTC iterations. While we agree that WER is the standard metric, CER provides additional insights into frame-level improvements achieved by our approach.
> >
> > ---
> >
> > Continued in the next comment ...

---

> > > ### Author Response · Authors · 2024-11-19
> > > **Official Comment by Authors (3/3)**
> > >
> > > **We appreciate your detailed feedback and constructive suggestions, which have helped us improve the quality of the paper.**
> > > We hope that the clarifications and planned modifications address your concerns. As the first method to unify ASR and TTS within a non-autoregressive framework, our work demonstrates state-of-the-art TTS performance and competitive ASR performance, which we believe constitutes a significant contribution to the field.
> > >
> > > ---
> > >
> > > **References**
> > > 1. R. Fan, W. Chu, P. Chang, and J. Xiao, "CASS-NAT: CTC Alignment-Based Single Step Non-Autoregressive Transformer for Speech Recognition," ICASSP 2021.
> > > 2. Microsoft Speech Service Evaluation, available at: [link](https://learn.microsoft.com/en-us/azure/ai-services/speech-service/how-to-custom-speech-evaluate-data?pivots=speech-studio#example-scenario-outcomes).
> > > 3. Chang, X., Yan, B., Fujita, Y., Maekaku, T., Watanabe, S. (2023) Exploration of Efficient End-to-End ASR using Discretized Input from Self-Supervised Learning. Proc. INTERSPEECH 2023.

---

> > > > ### Author Response · Authors · 2024-11-25
> > > > **Follow-Up on Discussion**
> > > >
> > > > Dear Reviewer D6js,
> > > >
> > > > Thank you once again for your thoughtful feedback, which we have carefully addressed in our rebuttal. We have incorporated the requested clarifications and additional evaluations into the updated paper, as outlined in our response.
> > > >
> > > > If you have any further questions or suggestions, we would be happy to address them before the discussion period concludes. Your insights have been invaluable in improving our work, and we sincerely appreciate the time and effort you have dedicated to reviewing our submission. We look forward to hearing your thoughts on the revised version.
> > > >
> > > > Best regards

---

> ### Comment · Reviewer_D6js · 2024-11-26
>
> Thanks for providing the explanations and revision plan. I still have several concerns on the paper.
>
> 1. I am thinking about the benefits of unifying the ASR and TTS tasks in one framework. After reading the MAS explanations, I just realized that one of the roles of CTC loss here is to provide alignment for TTS as a replacement of external alignment. Hence, it looks like the main task here would still be the TTS task. From my experience, CTC alignment is not always accurate, and it should be worse than the external alignment, e.g. obtained from hybrid models. This is why I was asking a comparison to using external alignment. On the other hand, it is very hard to see if the TTS task can help the ASR task as there is no such ablation (ASR loss only) and probably would not. I would be very cautious about the motivation and the storytelling of unifying ASR and TTS in the framework, especially an emphasize on the competitive ASR performance.
>
> 2. There is many non-autoregressive ASR work based on refining CTC outputs, some of which have similar ideas, e.g.
> - Align-Refine: Non-Autoregressive Speech Recognition via Iterative Realignment
> - Relaxing the Conditional Independence Assumption of CTC-based ASR by Conditioning on Intermediate Predictions
> especially for the second paper, concatenating CTC logits embedding with intermediate representations.
> These papers are not well referenced. This is also why I was mentioning CTC error correction is not that novel in the paper.
>
> 3. For comparison of latency, did you consider the speech discretization time for ASR and codec decoder time for TTS? If yes, please mention it or ignore if you already did it. If no, please add this time to make a fair comparison especially for ASR when comparing with non-discrete system in Table 9.
>
> Overall, I can raise my score to 5 but I cannot give a high value to the paper. The paper can be a good candidate for other conferences but should be lower to the acceptance broadline for ICLR in my mind.

---

> > ### Author Response · Authors · 2024-11-26
> > **Response to Follow-Up Concerns (1/3)**
> >
> > **Concern 1 (part a): Unified Framework for ASR and TTS: Motivation, Contributions, and Balance**
> > While it is true that TTS benefits directly from CTC alignment, the unification of ASR and TTS serves broader goals, including efficiency, generalizability, and scalability. Instead of positioning our work as “applying multi-task learning to improve individual tasks,” we position it as “the first work to unify the ASR and TTS tasks in a NAR framework by utilizing multi-task learning and without using external alignments.” Given this, the ASR and TTS tasks are integral to our framework.  We detail some of our key points below:
> >
> > - **Efficiency:**
> >   Unifying ASR and TTS reduces redundancy in maintaining separate models and training pipelines for related tasks. This approach improves both training and storage efficiency. Furthermore, the non-autoregressive (NAR) formulation ensures faster inference, making the framework practical for real-world use.
> >
> > - **Generalizability and Future Directions:**
> >   The shared latent space enables potential applications such as joint training, semi-supervised learning, and self-training, which are challenging to achieve with isolated ASR or TTS systems. While these are beyond the scope of this work, they highlight the broader vision of unifying related tasks.
> >
> > - **Clarified Storytelling:**
> >   Our claims focus on the practical benefits of unification rather than overstating ASR performance. Specifically:
> >   - T2V2 is a novel NAR framework for ASR and TTS using discrete speech tokens.
> >   - It introduces self-contained alignment via MAS for TTS and a CTC-correction mechanism for ASR improvement.
> >   - Experimental results validate state-of-the-art TTS performance and competitive ASR performance within the unified framework.
> >
> > - **Balanced Contributions and Demonstration of ASR-Only Performance:**
> >   The discrete conformer-CTC baseline in Table 9 represents our model trained with only the ASR loss (i.e., turning off all other losses). From the ablation study in Table 6, we observe that while the TTS task slightly harms the CTC predictions, our unified model—with the help of the CTC-correction mechanism and iterative refinement—achieves competitive performance compared to an ASR-only trained model. This demonstrates that the unified framework does not significantly compromise ASR performance and that ASR is not a secondary focus.
> >
> > Our unified framework highlights the feasibility of jointly addressing ASR and TTS tasks without compromising core performance metrics. It emphasizes efficiency and scalability while opening avenues for future research.
> >
> > **Concern 1 (part b): CTC Alignment as a Replacement for External Alignment**
> > You correctly observed that CTC loss provides alignment for the TTS task, replacing the need for external alignment tools. While tools such as Montreal Forced Aligner (MFA) might offer more accurate alignments in some scenarios, they come with several points that make them unsuitable for our unified framework, such as:
> >
> > - **Dependency on Phoneme-Level Alignment:**
> >   Tools like MFA operate at the phoneme level, requiring a grapheme-to-phoneme (G2P) model to map text to phonemes. This would complicate our pipeline, designed to work on text with minimal preprocessing. Introducing G2P adds unnecessary complexity and diverges from our goal of maintaining a streamlined framework.
> >
> > - **Potential issues with Token Frame Shifts:**
> >   MFA typically uses a 10ms frameshift, whereas our tokens are at a 20ms frameshift. This discrepancy introduces potential ambiguity when aligning MFA outputs to our token sequence, though the impact could be minimal.
> >
> > - **Storage and Workflow Complexity:**
> >   One of our key motivations for using discrete tokens is to achieve efficient storage by extracting and storing only the content and acoustic tokens. In contrast, external aligners like MFA require access to the raw waveform. While it is theoretically possible to convert our acoustic tokens back to the waveform for alignment purposes, this process is neither straightforward nor efficient. It introduces unnecessary complexity, undermining our design philosophy of simplicity and efficiency.
> >
> > Given these challenges, we prioritize a self-contained alignment mechanism via CTC. As shown in our supplementary material and various TTS evaluations, CTC-derived alignments are sufficient for producing intelligible and natural TTS outputs.
> >
> > We point to a recent comparison of forced alignment methods for readers interested in a broader analysis of alignment methods [1].
> >
> > [1]  Rousso, R., Cohen, E., Keshet, J., Chodroff, E. (2024). Tradition or Innovation: A Comparison of Modern ASR Methods for Forced Alignment. Proc. Interspeech 2024, 1525-1529, doi: 10.21437/Interspeech.2024-429.

---

> > > ### Author Response · Authors · 2024-11-26
> > > **Response to Follow-Up Concerns (2/3)**
> > >
> > > **Concern 2: CTC Error Correction and Related Work**
> > > We appreciate the reviewer pointing out these relevant works, and we sincerely apologize for not referencing them explicitly in our paper. While we included a discussion of [1], which covers both Align-Refine and Self-Conditioned CTC (second paper), we will expressly cite and discuss these works in the revised manuscript to ensure proper attribution and clarity.
> > >
> > > Regarding the similarities and distinctions with our method:
> > >
> > > - **Relation to Align-Refine:**
> > >   - We acknowledge the conceptual similarity, as both methods utilize an iterative refinement paradigm.
> > >   - However, our approach differs significantly in its formulation:
> > >     - Align-Refine employs a separate non-causal transformer decoder for refinement, integrating encoder features via cross-attention.
> > >     - In contrast, we take a more direct approach by treating refinement as a separate task. We reuse the same backbone encoder and directly combine intermediate CTC outputs with speech tokens for refinement, avoiding the need for additional modules such as decoders or cross-attention layers.
> > >
> > > - **Relation to Self-Conditioned CTC:**
> > >   - Similar to Self-Conditioned CTC, we utilize intermediate CTC outputs. However, there are key differences:
> > >     - Self-conditioned CTC applies intermediate CTC logit embeddings via additional linear projection layers at intermediate encoder stages, making it well-suited for dedicated ASR models.
> > >     - In our multi-task scenario, where the backbone supports both ASR and TTS tasks, adding such intermediate conditioning might not generalize well. Our approach maintains a lightweight design by avoiding additional linear layers or intermediate conditioning.
> > >     - Furthermore, Self-Conditioned CTC does not support iterative refinement, which is a central aspect of our method.
> > >
> > > We thank the reviewer for highlighting this and have revised the manuscript to include explicit citations and a detailed discussion of these works in the Appendix. This clarification will help position our contributions more effectively within the existing literature.
> > >
> > > [1] Higuchi, Yosuke, et al. "A comparative study on non-autoregressive modelings for speech-to-text generation." 2021 IEEE Automatic Speech Recognition and Understanding Workshop (ASRU). IEEE, 2021.
> > >
> > > **Concern 3: Latency Comparison**
> > > Yes, the inference runtimes for ASR and TTS include the speech discretization time for ASR, as well as codec decoding and prompt discretization time for TTS.  The only exception is IR-t2c (left-most column in Table 4), as it compares only the text-to-content token stage. Similarly, for the WhisperSpeech baseline, we measure runtimes only for the corresponding stage to ensure fair comparison.

---

> > > > ### Author Response · Authors · 2024-11-26
> > > > **Response to Follow-Up Concerns (3/3)**
> > > >
> > > > We appreciate your detailed feedback and constructive comments, which have helped us better position and clarify our work. Throughout the rebuttal, we have addressed your concerns, including clarifying the benefits of unifying ASR and TTS tasks, the motivations and contributions of our method, and improving the discussion of related work.
> > > >
> > > > We believe that our work makes a meaningful contribution as the first unified NAR framework for ASR and TTS operating on discrete speech tokens. This contribution has implications for task efficiency, scalability, and broader applications, and we hope it resonates with the broader ICLR audience. While we respect your perspective on the paper, we remain confident in its value and appreciate the opportunity to present our ideas for further feedback and discussion.

---

### Official Review · Reviewer_pcMe · 2024-11-04

**Soundness:** 2
**Presentation:** 3
**Contribution:** 3
**Rating:** 6
**Confidence:** 3

**Summary:**

The paper introduces a unified non-autoregressive model designed to perform named T2V2 for both of ASR and TTS task. The backbone of T2V2 is Conformer with RoPE which operates on discrete speech tokens. The model achieves the tasks the it's designed for by applying CTC for ASR and conditional masked LM approach for TTS, along with auxiliary tasks like CTC error correction and nconditional
speech masked LM. With extensive experiment and evaluation, the author achieves competitive TTS and ASR performance. The main contributions claimed in the paper include:
* The paper proposes the first unified framework based on discrete speech tokens handling ASR and TTS task.
* The introduction of CTC error correction auxiliary task potentially addresses conditional independence limitation in CTC objective and improves ASR performance.
* The MAS-based alignment solution is innovative and allows for effective TTS alignment without external alignment tools.

**Strengths:**

* The paper presents a unified framework capable of both ASR and TTS, which may streamline resource use and advance multitask learning for speech models.
* The proposed auxiliary tasks such as CTC error correction and unconditional speech MLM demonstrate a solution for future work to address challenges in multitask learning.
* The paper provides comprehensive experimental results include some ablation studies to justify the design choice and demonstrate T2V2's effectiveness.

**Weaknesses:**

* There is lack of discussions about or comparison with other multi-task models equipped with ASR & TTS capabilities such as [1] [2], etc. The discussion of related work isn't sufficient enough.
* The author states that the motivation of adopting non-autoregressive approach is the superior decoding time, however, there is lack of comparison of inference efficiency compared with its autoregressive counterparts.
* Presentation needs to be improved: for example: there is a mixed terminologies of "CTC token" "Text token" "Content token" "Acoustic token" used in Figure 1 and section 2/3, there should be a clear definition of the core terms used in the paper.



[1] Toyin, Hawau Olamide, Hao Li, and Hanan Aldarmaki. "STTATTS: Unified Speech-To-Text And Text-To-Speech Model." arXiv preprint arXiv:2410.18607 (2024).

[2] Yang, Runyan, et al. "PolySpeech: Exploring Unified Multitask Speech Models for Competitiveness with Single-task Models." arXiv preprint arXiv:2406.07801 (2024).

**Questions:**

1. What's the rational of doing the token length prediction by feeding text embedding through conformer backbone? This is odd because this is only task/step that conformer has to directly deal with text embedding and there seems to be risk of confusing the model.
2. The length prediction loss is not included in the final training objective in eqs 16, how is the length prediction head trained?
3. When the paper introduces CTC error correction task, it is stated in section 2.4.1 that the purpose is to address the independence assumption in CTC loss. From the mechanism of CTC error correction, it doesn't seem like the training breaks such assumption because the CTC logits was already detached from the computational graph. The author should elaborate more how this method essentially improves the performance. And how is it different from simply masking ground truth text embedding (or randomly augmented text) and do error correction.

---

> ### Author Response · Authors · 2024-11-19
> **Official Comment by Authors (1/1)**
>
> **We thank the reviewer for the detailed feedback and constructive questions. We are encouraged by the recognition of the strengths of our work.**
> Below we first list the revision plan based on the queries and suggestions of the reviewer, followed by detailed responses to each query.
>
> ---
>
> **Revision Plan**
> (Please check detailed responses for more information):
> 1. Include references with other AR multi-task models **[1]** and **[2]**.
> 2. Add inference runtime measurements and comparisons.
> 3. Update Figure 1 and unify the terminology.
> 4. Update Equation 16 to include the length prediction objective.
> 5. Include a discussion about the CTC correction task and the independence assumption.
>
> **NOTE**: The changes are colored in **blue** text in the revised manuscript for easy visibility.
>
> ---
>
> **Weaknesses**
>
> **Comparison with Other Multi-Task Models ([1] and [2])**
> Thank you for pointing out these references. [1] was posted on October 24, after the ICLR submission deadline, and we were not aware of  [2] as it is a preprint without many citations. Both works, however, focus on autoregressive (AR) models, whereas our work explores a non-autoregressive (NAR) approach.
>
> Additionally, we have already included citations for AR-based unified models in the manuscript, we will add these references as well for completeness.
>
> ---
>
> **Inference Efficiency Comparison**
> We acknowledge the need for inference time comparisons with AR models, and we did not originally include these since it is well-known that NAR models are faster than AR models for inference. However, we now realize that it is important to demonstrate the inference efficiency our approach explicitly, and will include inference runtime measurements to the revised manuscript.
>
> Please also refer to the “Inference Runtimes” comment at the top.
>
> ---
>
> **Mixed Terminology**
> We agree that the use of mixed terminologies (e.g., "CTC token," "Text token," "Content token," "Acoustic token") in Figure 1 and Sections 2/3 could confuse readers. We will unify the terminologies as well as simplify Figure 1 for easier understanding.
>
> ---
>
> **Questions**
>
> **Rationale for Length Prediction Using Text Embeddings**
> While we recognize the potential risk of confusing the model, our rationale is that text embeddings are simply another representation of the same underlying language, albeit at a different and irregular sampling frequency., but with similar linguistic patterns, which enables the model to generalize across tasks effectively.
>
> The length prediction task, as shown in our experiments, does not hamper optimization, and the model achieves competitive performance across tasks.
>
> ---
>
> **Length Prediction Loss in Final Training Objective**
> You are correct that the length prediction loss was inadvertently omitted in Eq. 16. The length prediction objective is also added to the overall objective and optimized together with all other tasks. We will update the equation to include the length prediction objective.
>
> ---
>
> **CTC Error Correction and Independence Assumption**
> The initial pass of CTC follows the original independence assumption. However, in subsequent passes, the intermediate CTC predictions (including blanks) are masked based on confidence (log probabilities) and combined with speech tokens. The output tokens are predicted without reduction, allowing the model to access partial text contexts.
>
> This iterative refinement enables the model to relax the independence assumption and utilize the interdependencies among text predictions in subsequent passes, which improves performance.
>
> Furthermore, Unlike approaches like Mask-CTC[3] and Improved-Mask-CTC[4], which operate on reduced intermediate CTC outputs and explicitly handle substitutions, insertions, and deletions, our method operates on un-reduced outputs, allowing implicit handling of all error types.
>
> ---
>
> **Thank you for recognizing the contributions of our work and for your valuable feedback. We look forward to incorporating these improvements to strengthen the manuscript.**
>
> ---
>
> **References**
> 1. Toyin, Hawau Olamide, Hao Li, and Hanan Aldarmaki. "STTATTS: Unified Speech-To-Text And Text-To-Speech Model." arXiv preprint arXiv:2410.18607 (2024).
> 2. Yang, Runyan, et al. "PolySpeech: Exploring Unified Multitask Speech Models for Competitiveness with Single-task Models." arXiv preprint arXiv:2406.07801 (2024).
> 3. Higuchi, Yosuke, et al. "Mask CTC: Non-autoregressive end-to-end ASR with CTC and mask predict." arXiv preprint arXiv:2005.08700 (2020).
> 4. Higuchi, Yosuke, et al. "Improved Mask-CTC for non-autoregressive end-to-end ASR." ICASSP 2021-2021 IEEE International Conference on Acoustics, Speech and Signal Processing (ICASSP). IEEE, 2021.

---

> > ### Author Response · Authors · 2024-11-25
> > **Follow-Up on Discussion**
> >
> > Dear Reviewer pcMe,
> >
> > Thank you once again for your thoughtful feedback, which we have carefully addressed in our rebuttal. We have incorporated the requested clarifications and additional evaluations into the updated paper, as outlined in our response.
> >
> > If you have any further questions or suggestions, we would be happy to address them before the discussion period concludes. Your insights have been invaluable in improving our work, and we sincerely appreciate the time and effort you have dedicated to reviewing our submission. We look forward to hearing your thoughts on the revised version.
> >
> > Best regards

---

### Official Review · Reviewer_quVm · 2024-11-04

**Soundness:** 2
**Presentation:** 3
**Contribution:** 3
**Rating:** 8
**Confidence:** 4

**Summary:**

This paper addresses the challenge of unifying automatic speech recognition (ASR) and text-to-speech synthesis (TTS) using discrete speech tokens derived from self-supervised models. The authors introduce a CTC error correction task to enhance ASR performance by refining raw CTC outputs. Additionally, they propose a monotonic alignment search (MAS) technique with intermediate CTC outputs to achieve TTS alignment without relying on external tools. Experiments were conducted on the LibriHeavy dataset, with both objective and subjective evaluations of the synthesized speech.

**Strengths:**

* This paper explores a challenging problem that enables the unification of ASR and TTS tasks, allowing the framework to benefit from the underlying complexities of each task.
* The claims were tested, demonstrating improvements in TTS performance as intended, while achieving comparable performance for the ASR task.

**Weaknesses:**

It is unclear how much each task contributed to the overall improvement across tasks. Additionally, the performance of the proposed model on different attributes—such as pitch, volume, and speed of synthesized speech—has not been clearly detailed.

**Questions:**

* (Lines 47-49): "While multitask learning offers potential for parameter sharing, the performance of individual tasks might suffer due to the inherent complexity of training multiple tasks simultaneously." — The authors should provide references to prior work to support this claim. Additionally, if multitask learning poses such challenges, how were the authors able to handle both text-to-speech and speech-to-text effectively in the proposed unified model?

* Simplifying Figure 1 would make it easier for readers to understand the proposed framework.

* (Line 64): Why was the Conformer architecture with rotary positional embeddings (RoPE) chosen? Could the authors expand on the reasoning for using RoPE? Does RoPE offer benefits for both ASR and TTS tasks?

* (Line 261): It is unclear how the authors optimized the individual losses within their multitask learning framework. Was any weighting applied to the individual losses, and how did each loss contribute to the optimization of each task?

* There is some confusion regarding the dataset subsets used to train the unified model. In lines 311-315, the authors mention a 509-hour subset of LibriHeavy, while in lines 484-485 they refer to training a larger version of the model on a 50K-hour LibriHeavy subset for discrete ASR. What is the relationship between these two training stages? In the second ASR training, is the model initialized from the first training stage, or is it a separately trained ASR model?

* In Table 8, could the authors include baselines for the discrete Zipformer-CTC model for completeness?

* How does the proposed unified model perform on other TTS attributes, such as pitch, volume, and speed?

---

> ### Author Response · Authors · 2024-11-19
> **Official Comment by Authors (1/2)**
>
> **We thank the reviewer for the detailed feedback and constructive questions. We are encouraged by the recognition of the strengths of our work.**
>
> Below, we first list the revision plan based on the queries and suggestions of the reviewer, followed by detailed responses to each query.
>
> ---
>
> **Revision Plan**
> (Please check detailed responses for more information):
> 1. Simplify Figure 1.
> 2. Include references and discussion about multi-task learning challenges and optimization of losses.
> 3. Include references and discussion about the choice of Conformers with RoPE for ASR and TTS tasks.
> 4. Include clarification about training from scratch on the large subset for the scaling experiment.
> 5. If possible, include preliminary results of the discrete Zipformer-CTC baseline.
>
> **NOTE**: The changes are colored in **blue** text in the revised manuscript for easy visibility.
>
> ---
>
> **Weaknesses**
>
> **Contribution of Each Task to Overall Performance**
> Instead of positioning our work as “applying multi-task learning to improve individual tasks,” we position it as “the first work to unify the ASR and TTS tasks in a NAR framework by utilizing multi-task learning and without using external alignments.” Given this, the ASR and TTS tasks are integral to our framework.
>
> While the ASR task could be trained on its own (basically our Conformer-CTC baseline), the TTS task cannot be trained without the aid of the ASR task, as the alignment is deeply linked to the ASR task. Regarding the auxiliary tasks, the speech MLM and CTC correction tasks aim to improve the performance of the respective TTS and ASR tasks. We performed ablation studies to analyze the effects of these tasks and the results are in Tables 1 and 6. These show that the auxiliary tasks help improve individual task performances.
>
> ---
>
> **Analysis of TTS Attributes (Pitch, Volume, Speed)**
> The Soundstorm model influences the pitch and volume of the synthesized speech, thus that is not the focus of our work. As such, analyzing these attributes would not directly reflect the advantages or disadvantages of T2V2. Regarding the speaking rate (speed), it relies on the token-wise length prediction task, once the lengths are predicted, it is possible to scale the speed by multiplying each length with a factor, to make it faster or slower, similar to how it is done in prior work like VITS **[1]**.
>
> ---
>
> **Questions**
>
> **Multitask Learning Challenges and Optimization of Losses**
> Multitask learning, particularly multi-objective optimization (MOO), is known to pose challenges, such as balancing performance across tasks, especially for conflicting objectives. Prior works, such as [2], detail these challenges and common approaches like Pareto optimality, genetic algorithms, and scalarization.
>
> Our framework adopted the simplest weighted-sum scalarization approach, assigning equal weights to all task-specific losses. This choice was based on the assumption that the tasks are cooperative rather than conflicting, especially in our single-language scenario. The equal weighting reflects the equal importance of the tasks in our unified model and the experimental results demonstrate that this simple equally weighted scalarization helps achieve state-of-the-art performance.
>
> ---
>
> **Simplification of Figure 1**
> We acknowledge that the figure is a bit complicated and will update it in the revised manuscript.
>
> ---
>
> **Choice of Conformer with RoPE**
> The Conformer architecture is well-suited for speech processing tasks, capturing both local and global dependencies. Adding Rotary Positional Embeddings (RoPE) enhances its ability to handle variable-length sequences, benefiting both ASR and TTS. Conformers with RoPE have demonstrated effectiveness in ASR [3] and TTS [4].
>
> ---
>
> Continued in the next comment ...

---

> > ### Author Response · Authors · 2024-11-19
> > **Official Comment by Authors (2/2)**
> >
> > **Optimization of Individual Losses**
> > We used equal weighting for all tasks during training, converting the MOO problem to a Single Objective Optimization (SOO) via scalarization. This choice reflects the equal importance of all tasks in our framework.
> >
> > ---
> >
> > **Dataset Subsets and Training**
> > The 509-hour “libriheavy-small”  is a subset of the 50K-hour “libriheavy-large” dataset. As for the scaling experiment, we trained the models from scratch on the 50K-hour subset and did not initialize from the small training. Moreover, we trained with the full multi-task objective, and not just the ASR task.
> >
> > ---
> >
> > **Baselines for Discrete Zipformer-CTC**
> > We will attempt to implement the discrete Zipformer-CTC architecture and conduct experiments to include its results in Table 8. However, fully implementing  the model might not be feasible within the rebuttal timeline. For the results presented in the paper, we utilized the pre-trained model provided by the original authors. Modifying this architecture for discrete-CTC experiments within their setup is non-trivial, and porting it to our framework will require additional time and effort. We will do our best to include preliminary results in the revision.
> >
> > ---
> >
> > **We appreciate the thoughtful review and will incorporate these improvements to strengthen the manuscript. Thank you for your constructive feedback.**
> >
> > ---
> >
> > **References**
> > 1. Kim, Jaehyeon, Jungil Kong, and Juhee Son. "Conditional variational autoencoder with adversarial learning for end-to-end text-to-speech." International Conference on Machine Learning. PMLR, 2021.
> > 2. L. Xiujuan and S. Zhongke, "Overview of multi-objective optimization methods," in Journal of Systems Engineering and Electronics, vol. 15, no. 2, pp. 142-146, June 2004.
> > 3. Li, Shengqiang, Menglong Xu, and Xiao-Lei Zhang. "Conformer-based end-to-end speech recognition with rotary position embedding." 2021 Asia-Pacific Signal and Information Processing Association Annual Summit and Conference (APSIPA ASC). IEEE, 2021.
> > 4. Borsos, Zalán, et al. "Soundstorm: Efficient parallel audio generation." arXiv preprint arXiv:2305.09636 (2023).

---

> > > ### Comment · Reviewer_quVm · 2024-11-23
> > > **Response to Authors**
> > >
> > > Thank you for your additional analysis. The insights you provided are valuable and effectively highlight the advantages of a unified non-autoregressive automatic speech recognition and text-to-speech model. I suggest that the authors further refine this analysis, for instance, by elaborating on how implicit CTC error correction improves all error types—substitutions, insertions, and deletions—through a detailed breakdown of these improvements. Incorporating this into the final version would improve the manuscript. I am happy to improve my score.

---

> > > > ### Author Response · Authors · 2024-11-23
> > > > **Response to Reviewer Comment**
> > > >
> > > > Thank you for your thoughtful feedback and positive remarks on our unified non-autoregressive ASR and TTS model. We greatly appreciate your suggestion to further refine the analysis by elaborating on how our CTC error correction implicitly improves substitutions, insertions, and deletions. Following your recommendation, we have incorporated the following enhancements into the manuscript:
> > > >
> > > > 1. **Description of implicit error type handling and differentiation from prior work**:
> > > >    - In **Section 2.4.1**, we included a detailed explanation of how our implicit CTC error correction mechanism handles various error types.
> > > >    - We also distinguished our method from prior works like **Mask-CTC** and **Improved-Mask-CTC**, emphasizing the novel aspects of our approach.
> > > >
> > > > 2. **New Table for Error Breakdown**:
> > > >    - We added **Table 8** to present a detailed breakdown of error types—substitutions, insertions, and deletions. This table highlights consistent improvements across all error types with our proposed correction mechanism.
> > > >    - For clarity, we included a corresponding description in **Section 3.5.1**, explaining these results and their significance.
> > > >
> > > > Below is the new table included in the manuscript for your reference:
> > > >
> > > > |          | Substitutions | Insertions | Deletions |
> > > > |----------|---------------|------------|-----------|
> > > > | **w/o CORR** | 1.300         | 0.090      | 0.140     |
> > > > | **w CORR**   | **1.255** (*↓ 3.46%*)     | **0.082** (*↓ 8.89%*)  | **0.135** (*↓ 3.57%*) |
> > > >
> > > >
> > > > These additions underscore how our correction mechanism provides consistent and balanced improvements across all error types, showcasing its robustness. We believe these refinements address your suggestion and enhance the clarity and impact of the manuscript.
> > > >
> > > > Thank you once again for your constructive feedback and willingness to improve your score. Your guidance has been instrumental in strengthening our work.

---

> > > > > ### Comment · Reviewer_quVm · 2024-11-24
> > > > > **Response to Authors**
> > > > >
> > > > > Thank you for providing the detailed analysis. Overall, I have revised my score after careful consideration of the responses.

---

> > > > > > ### Author Response · Authors · 2024-11-25
> > > > > > **Thank You**
> > > > > >
> > > > > > Dear Reviewer quVm,
> > > > > >
> > > > > > We sincerely thank you for your thoughtful feedback and for taking the time to review our rebuttal and update your evaluation. Your insights and suggestions have been invaluable in helping us improve our manuscript. We deeply appreciate your engagement and the effort you’ve put into reviewing our work.
> > > > > >
> > > > > > Best regards

---

### Author Response · Authors · 2024-11-19
**Inference Runtimes**

**Inference Runtime Comparisons**

We conducted inference runtime comparisons as follows:

**TTS**: Time taken to generate speech for a 405-character text input averaged over 100 trials on an H100 GPU

**ASR**: Time taken to recognize a 60s speech input averaged over 100 trials on an H100 GPU

**Results:**

**Inference runtimes for the text-to-content tokens stage.**
| Model                                | Runtime       |
|:-------------------------------------|:--------------|
| WhisperSpeech (AR)                        | 2.842 ± 0.007 |
 | Ours (iter=1)       | 0.013 ± 0.000 |
| Ours (iter=4)       | 0.033 ± 0.000 |
| Ours (iter=4; CFG) | 0.057 ± 0.000 |

**Inference runtimes for end-to-end text to speech waveform.**
| Model                                | Runtime        |
|:-------------------------------------|:---------------|
| StyleTTS2                            | 0.268 ± 0.003  |
| YourTTS                              | 0.110 ± 0.004  |
| HierSpeechPP                         | 0.164 ± 0.004  |
| XTTS_v2                              | 2.597 ± 0.029  |
| WhisperSpeech                        | 17.911 ± 0.042 |
| Ours (iter=1)       | 0.528 ± 0.001  |
| Ours (iter=4)       | 0.546 ± 0.000  |
| Ours (iter=4; CFG) | 0.568 ± 0.000  |

**Inference runtimes for ASR and number of correction iterations.**

| Iterations                                | Runtime        |
|:-------------------------------------|:---------------|
| w/o CORR                            | 0.32 ± 0.02  |
| 1                              | 0.40 ± 0.03  |
| 4                         | 0.39 ± 0.03  |
| 8                              | 0.42 ± 0.03  |
| 16                        | 0.47 ± 0.03 |
| 32                        | 0.60 ± 0.03 |

**Inference runtimes for ASR compared with baselines.**

| Model                                | Runtime        |
|:-------------------------------------|:---------------|
| Zipformer-transducer-small                            | 1.49 ± 0.07 |
| Zipformer-transducer-large                             | 1.51 ± 0.14  |
| Conformer-CTC-small                         | 0.32 ± 0.02  |
| Ours-small                              | 0.47 ± 0.03  |
| Conformer-CTC-large                        | 0.34 ± 0.02 |
| Ours-large                        | 0.55 ± 0.02 |

We have added these in the coressponding tables in the updated paper.

---

### Author Response · Authors · 2024-11-19
**Summary of Changes in Revised Paper**

**Summary of Changes in Revised Paper**
- Slight rewording of the claims. (Sec. 1)
- Update and simplify the figure. Separated the individual tasks to clearly show the input-output relationships without complicated interlocking arrows. Color-coded and added legend for the various inputs, outputs, and intermediate outputs. (Fig. 1)
- Add inference runtime measurements and comparisons. (Tables 4, 7, and 9)
- Include references to other AR multi-task ASR-TTS models. (Sec. 1)
- Include additional CTC correction references. (Sec. 1)
- Include a discussion on the advantages of discrete ASR. (Sec. 1)
- Include references and discussion about choosing Conformers with RoPE for ASR and TTS tasks. (Sec. 2.2)
- Include details on how MAS is applied to our framework. (Sec. 2.3.2)
- Include a discussion about the CTC correction task and the independence assumption and positioning of our CTC correction formulation's novelty compared to Mask-CTC and Improved Mask-CTC and relation to Align-Refine and Self-conditioned-CTC. (Sec. 2.4.1)
- Include references and discussion about multi-task learning challenges and optimization of losses. (Sec. 1 and Sec. 2.6)
- Update the final loss to include the length prediction objective. (Eq. 16)
- Added Table 8 to present a detailed breakdown of error types—substitutions, insertions, and deletions. This table highlights consistent improvements across all error types with our proposed correction mechanism and updated the description. (Sec. 3.5.1, Table 8)
- Include clarification about training from scratch on the large subset for the scaling experiment. (Sec. 3.5.2)
- Include a discussion about marginal performance degradation due to the SMLM task in single-pass inference, and how iterative inference with CFG (enabled by the SMLM task) significantly boosts CER for TTS and the CORR task compensates for it in ASR. (Sec. 3.4.1 and 3.5.1)
- Add an appendix to further elaborate on the similarities and differences between our CTC correction and Align-refine and Self-conditioned-CTC. (Appendix A)

**NOTE**: The changes in the paper are colored in **blue** text for easy visibility.

---

### Meta-Review · Area_Chair_NmUW · 2024-12-20

**Metareview:**

The paper proposes a non-autoregressive system for joint TTS and ASR. To enable training of both tasks, CTC alignments, discrete tokens, and various auxiliary losses are used.

I recommend an acceptance for its potential improve both ASR and TTS, especially for efficient inference.

However, I do want to remind the authors that all reviewers unanimously agreed that there is a lack of proper attribution of the improvements. In simple terms, we do not know which parts of the system are necessary and which parts of the system are good-to-have's. The ablation study in Table 8 does not answer this fundamental question.

Reviewers pcMe, D6js, and AXxb raised issues about writing. Issues include organization, vague descriptions, and imprecise use of terms, hampering reproducibility. Making significant changes to the original submission and patching things here and there during a short period of time is typically discouraged. I'd recommend the authors to carefully re-think the organization and contribution of the work.

I want to highlight one change that are not yet implemented, an issue raised by Reviewer pcMe. The use of "content tokens" and "acoustic tokens" is problematic, and is perhaps best to call them HuBERT tokens and DAC tokens to avoid ambiguity and misunderstanding. In particular, HuBERT tokens carry not just "content" (linguistic content I suppose) but many other information. See the two papers below.

* Yeh and Tang, Estimating the Completeness of Discrete Speech Units, 2024
* Choi et al., Self-Supervised Speech Representations are More Phonetic than Semantic, 2024

Coupled with text tokens, it becomes difficult to keep track of the types (in the sense of programming language) of all the input and output, intermediate or not. It's probably best to explicit detail the set of values that a variable can be. For example, HuBERT tokens should have values in $\\{1, ..., 1024\\}$ and when they are embedded they should be vectors in $\mathbb{R}^d$.

Reviewers have provided valuable feedback, and I encourage the authors to consider them for improving the paper.

**Additional Comments On Reviewer Discussion:**

A lot of the discussion had been around improving the exposition of the proposed method. It was unclear if some of the improvements are within the margin of random noise. The authors had spent significant effort trying to convince the reviewers. The discussion was healthy, and the reviewers raised their scores.

---

### Decision · Program_Chairs · 2025-01-22

Accept (Poster)